# Score Neural Operator: A Generative Model for Learning and Generalizing Across Multiple Probability Distributions

## Abstract

Most existing generative models are limited to learning a single probability distribution from the training data and cannot generalize to novel distributions for unseen data. An architecture that can generate samples from both trained datasets and unseen probability distributions would mark a significant breakthrough. Recently, score-based generative models have gained considerable attention for their comprehensive mode coverage and high-quality image synthesis, as they effectively learn an operator that maps a probability distribution to its corresponding score function. In this work, we introduce the *Score Neural Operator*, which learns the mapping from multiple probability distributions to their score functions within a unified framework. We employ latent space techniques to facilitate the training of score matching, which tends to over-fit in the original image pixel space, thereby enhancing sample generation quality. Our trained Score Neural Operator demonstrates the ability to predict score functions of probability measures beyond the training space and exhibits strong generalization performance in both 2-dimensional Gaussian Mixture Models and 1024-dimensional MNIST double-digit datasets. Importantly, our approach offers significant potential for few-shot learning applications, where a single image from a new distribution can be leveraged to generate multiple distinct images from that distribution.

## 1 Introduction

Generative modeling has become a cornerstone of modern artificial intelligence and machine learning, with applications ranging from image synthesis to natural language processing Brown (2020); Karras et al. (2019). These models can be broadly categorized into two main approaches: likelihood-based methods, such as autoregressive models and normalizing flows Rezende & Mohamed (2015); Kingma & Dhariwal (2018); Papamakarios et al. (2021), and implicit generative models, exemplified by Generative Adversarial Networks (GANs) Goodfellow et al. (2014); Arjovsky et al. (2017). While these approaches have shown remarkable success, they often face limitations in terms of model architecture constraints or reliance on surrogate objectives for likelihood approximation Theis et al. (2015); Salimans et al. (2016); Papamakarios et al. (2021).

In recent years, score-based generative models have emerged as a powerful alternative, gaining considerable attention for their ability to produce high-quality samples and provide comprehensive mode coverage. These models operate by approximating the score function (the gradient of the log-density) of the target distribution through minimizing the Fisher divergence Song & Ermon (2019); Hyvärinen & Dayan (2005). Once trained, new samples can be generated using Langevin dynamics Song et al. (2020). The appeal of score-based methods lies in their ability to avoid the limitations of both likelihood-based and implicit generative models while maintaining high-quality output Song & Ermon (2019); Song et al. (2020); Ho et al. (2020).

However, most existing generative models, including traditional score-based approaches, are designed to learn and sample from a single probability distribution. This limitation becomes particularly apparent in scenarios where we need to generate samples from multiple related but distinct distributions, or when faced with novel, unseen distributions. The ability to learn and generalize

across multiple distributions would mark a significant advancement in the field, enabling more flexible and adaptable generative models.

To address this capability gap, we introduce the Score Neural Operator, a novel framework that learns to map multiple probability distributions to their corresponding score functions within a unified model. Our approach leverages recent advancements in operator learning Azizzadenesheli et al. (2024); Lu et al. (2019); Seidman et al. (2022) to capture the underlying relationships between different distributions and their score functions. By doing so, we enable the model to not only generate high-quality samples from trained distributions but also to generalize to unseen distributions without requiring retraining. The main contributions of our work can be summarized as follows:

- We introduce the Score Neural Operator, a novel architecture that learns to map multiple probability distributions to their score functions, enabling generalization to unseen distributions.

- We employ latent space techniques to facilitate score matching in high-dimensional spaces, improving both the quality of generated samples and the efficiency of the training process.

- We demonstrate the effectiveness of our approach on both low-dimensional Gaussian Mixture Models and high-dimensional image data (MNIST double-digit), showing strong generalization performance to unseen distributions.

- We showcase the potential of our method for few-shot learning applications, where a single image from a new distribution can be used to generate multiple distinct samples from that distribution.

Our work represents a significant step towards more flexible and generalizable generative models, with potential applications in transfer learning, few-shot generation, and adaptive AI systems.

## 2 RELATED WORK

**Score-based Generative Models.** Score-based generative models have gained prominence in recent years due to their ability to produce high-quality samples without the need for adversarial training or explicit density estimation. The foundational work by Song and Ermon Song & Ermon (2019) introduced the concept of score matching with Langevin dynamics for sample generation. This was further developed by Song et al. Song et al. (2020), who proposed a continuous-time formulation of the score-based generative process using stochastic differential equations (SDEs). Recent advancements in score-based models have focused on improving sample quality and generation speed. Notably, Vahdat et al. Vahdat et al. (2021) introduced a method to perform score matching in a lower-dimensional latent space using a variational autoencoder, significantly improving both the quality of generated samples and the efficiency of the training process. Rombach et al. Rombach et al. (2022) further extended this idea by incorporating cross-attention mechanisms, enabling conditioning on various types of embeddings. While these developments have greatly enhanced the capabilities of score-based models, they remain primarily focused on learning and sampling from a single distribution. Our work aims to address this limitation by learning a mapping across multiple distributions.

**Operator Learning.** Operator learning has emerged as a powerful paradigm for capturing mappings between function spaces, with applications in various scientific and engineering domains Kovachki et al. (2021). Notable approaches in this field include the Deep Operator Network (DeepONet) Lu et al. (2019), which can approximate nonlinear operators based on the universal approximation theorem, and the Fourier Neural Operator Li et al. (2020), which leverages spectral methods for efficient operator learning. Of particular relevance to our work is the NOMAD (Nonlinear Manifold Decoders for Operator Learning) framework introduced by Seidman et al. Seidman et al. (2022). NOMAD's ability to handle high-dimensional outputs and provide continuous representations makes it well-suited for our task of learning score functions across multiple distributions. While operator learning has found success in various scientific computing applications, its potential in generative modeling has been relatively unexplored. Our work bridges this gap by applying operator learning techniques to the domain of score-based generative models.

**Few-shot Learning in Generative Models.** Few-shot learning, the ability to learn from a small number of examples, has been a long-standing challenge in machine learning. In the context of generative models, few-shot learning is particularly challenging due to the need to capture complex data distributions from limited samples. Recent works have explored few-shot learning in generative models, such as the approach by Rezende et al. (2016) using meta-learning for one-shot generalization in generative models. However, these methods often struggle with generating diverse samples from limited data, frequently resulting in overfitting to the few available examples. Our Score Neural Operator aims to address this challenge by learning a generalizable mapping across distributions, enabling the generation of diverse samples even when presented with a single example from a new distribution. This capability sets our approach apart from existing few-shot generative methods and opens up new possibilities for adaptive and flexible generative modeling.

## 3 BACKGROUND

### 3.1 SCORE-BASED GENERATIVE MODEL

Recall that for score-based generative model the diffusion process is governed by a stochastic differential equation (SDE):

$$d\mathbf{x} = \mathbf{f}(\mathbf{x}, t)dt + \mathbf{G}(\mathbf{x}, t)d\mathbf{w}, \tag{1}$$

where $\mathbf{f}(\cdot, t) : \mathbb{R}^d \to \mathbb{R}^d$ and $\mathbf{G}(\cdot, t) : \mathbb{R}^d \to \mathbb{R}^{d \times d}$. At $t = 0$, $\mathbf{x}(0)$ is sampled from the original data distribution $p_0(\mathbf{x})$. Those input samples are disturbed by noise through a Stochastic Differential Equation equation 1, and evolve into a Gaussian prior with fixed mean and covariance at $t = 1$. Subsequently, a sample is drawn from this prior and denoised via the reverse diffusion process, which is structured as follows to reconstruct the original data distribution:

$$d\mathbf{x} = \left[\mathbf{f}(\mathbf{x}, t) - \nabla \cdot [\mathbf{G}(\mathbf{x}, t)\mathbf{G}(\mathbf{x}, t)^{\mathrm{T}}] - \mathbf{G}(\mathbf{x}, t)\mathbf{G}(\mathbf{x}, t)^{\mathrm{T}}\nabla_{\mathbf{x}} \log p_t(\mathbf{x})\right] dt + \mathbf{G}(\mathbf{x}, t)d\bar{\mathbf{w}}. \tag{2}$$

All the needed information to generate new samples is the score function $\nabla_{\mathbf{x}} \log p_t(\mathbf{x})$, which is approximated by a parameterized neural network $\mathbf{s}_\theta(\mathbf{x}(t), t)$. In Song et.al Song et al. (2020), the loss function is given by,

$$\min_\theta \mathbb{E}_{t \sim \mathcal{U}(0,T)}[\lambda(t)\mathbb{E}_{\mathbf{x}(0) \sim p_0(\mathbf{x})}\mathbb{E}_{\mathbf{x}(t) \sim p_{0t}(\mathbf{x}(t)|\mathbf{x}(0))}[\|\mathbf{s}_\theta(\mathbf{x}(t), t) - \nabla_{\mathbf{x}(t)} \log p_{0t}(\mathbf{x}(t) \mid \mathbf{x}(0))\|_2^2]], \tag{3}$$

where $\lambda(t)$ is the weighting function usually chosen to be inversely proportional to $\mathbb{E}[\|\nabla_{\mathbf{x}} \log p_{0t}(\mathbf{x}(t) \mid \mathbf{x}(0))\|_2^2]$, $p_{0t}$ is the transitional kernel that transforms the distribution $p_0(\mathbf{x})$ at time 0 into the distribution $p_t(\mathbf{x})$ at time $t$ and has a closed form by choosing specific $\mathbf{f}$ and $\mathbf{G}$.

Equation 3 targets the score function of a single distribution. We extend this training objective to multiple probability distributions by utilizing an operator learning architecture, which is adept at learning mappings between functional spaces. To achieve this, we require a method to embed a probability distribution into a vector such that it captures sufficient statistical information. A straightforward approach is to compute the characteristic function of the probability measure at predetermined points. However, the size of the embedding increases exponentially with the dimension, making this method computationally prohibitive for high-dimensional data, such as images. The Vector Quantized-Variational Autoencoder (VQ-VAE) Van Den Oord et al. (2017) employs an encoder to compress input samples into a discrete latent space, or codebook. However, different samples from the same distribution are mapped to distinct latent vectors in the codebook. Ideally, we desire that different samples from the same distribution yield the same embedding, as this would enhance the stability of the training process. Thus, an efficient and effective method for embedding probability distributions is essential. In the next section, we introduce two scalable approaches for embedding probability distributions as data dimensionality increases.

### 3.2 KERNEL MEAN EMBEDDINGS

Consider a set $\mathcal{X}$, and a kernel $k : \mathcal{X} \times \mathcal{X} \to \mathbb{R}$ which is positive definite and symmetric. Define the vector space of functions

$$\mathcal{H}_0 := \left\{\sum_{i=1}^n \alpha_i k(\cdot, x_i) \,\middle|\, n \in \mathbb{N}, \ \alpha_i \in \mathbb{R}, \ x_i \in \mathcal{X}\right\}.$$

We can give this space the inner product

$$\left\langle \sum_{i=1}^{n} \alpha_i k(\cdot, x_i), \sum_{i=1}^{m} \beta_i k(\cdot, y_i) \right\rangle = \sum_{i=1}^{n} \sum_{j=1}^{m} \alpha_i \beta_j k(x_i, y_j). \tag{4}$$

The completion of $\mathcal{H}_0$ with respect to the norm induced by the inner product equation 4 is the Reproducing Kernel Hilbert Space (RKHS) associated with the kernel $k$, denoted $\mathcal{H}_k$. It is a Hilbert space of functions from $\mathcal{X} \to \mathbb{R}$ with an inner product from equation 4.

An important property of $\mathcal{H}_k$ is the *reproducing property* which states that for any $f \in \mathcal{H}_k$ and any $x \in \mathcal{X}$,

$$f(x) = \langle f, k(\cdot, x) \rangle, \tag{5}$$

that is, pointwise evaluation is a linear functional on $\mathcal{H}_k$ (this is sometimes taken as part of the definition of an RKHS). For more information on RKHS's see Manton et al. (2015).

Let $\mathcal{P}(\mathcal{X})$ be the space of probability measures on a set $\mathcal{X}$. A kernel $k : \mathcal{X} \times \mathcal{X} \to \mathbb{R}$ (under some reasonable assumptions) induces a mapping from $\mathcal{P}(\mathcal{X}) \to \mathcal{H}_k$ defined by

$$\mathbb{P} \longmapsto \mu_{\mathbb{P}} := \int_{\mathcal{X}} k(\cdot, x) d\mathbb{P}(x), \tag{6}$$

where we call $\mu_{\mathbb{P}}$ the *kernel mean embedding* of $\mathbb{P}$. Note that $\mu_{\mathbb{P}} \in \mathcal{H}_k$ and is therefore a function $\mu_{\mathbb{P}} : \mathcal{X} \to \mathbb{R}$. In the case that $\mathbb{P}$ is an empirical distribution, that is

$$\hat{\mathbb{P}} = \frac{1}{n} \sum_{i=1}^{n} \delta_{x_i}, \tag{7}$$

we have that

$$\mu_{\hat{\mathbb{P}}} = \frac{1}{n} \sum_{i=1}^{n} k(\cdot, x_i). \tag{8}$$

The definition of the kernel mean embedding along with the reproducing property in $\mathcal{H}_k$ allows one to show that we may represent an inner product of kernel mean embeddings in $\mathcal{H}_k$ as an expectation of the kernel over the product measure of the associated distributions. Specifically, for any probability measures $\mathbb{P}, \mathbb{Q} \in \mathcal{P}(\mathcal{X})$,

$$\langle \mu_{\mathbb{P}}, \mu_{\mathbb{Q}} \rangle_{\mathcal{H}_k} = \int_{\mathcal{X}} \int_{\mathcal{X}} k(x, y) d\mathbb{P}(x) d\mathbb{Q}(y). \tag{9}$$

In the case that we consider two empirical distributions

$$\hat{\mathbb{P}} = \frac{1}{n} \sum_{i=1}^{n} \delta_{x_i}, \quad \hat{\mathbb{Q}} = \frac{1}{m} \sum_{i=1}^{m} \delta_{y_i},$$

we can compute the inner product of their embedding in $\mathcal{H}_k$ as

$$\langle \mu_{\hat{\mathbb{P}}}, \mu_{\hat{\mathbb{Q}}} \rangle_{\mathcal{H}_k} = \frac{1}{n} \frac{1}{m} \sum_{i=1}^{n} \sum_{j=1}^{m} k(x_i, y_j). \tag{10}$$

For more details about kernel mean embeddings see Muandet et al. (2012; 2017).

We then conduct Principal Component Analysis on $\mu_{\mathbb{P}_i}$ in RKHS to reduce dimensions. For any probability measure $\mathbb{P}_t$, the $k$-th element of $\mathbf{u}$ is given by,

$$u_k = \sum_{j=1}^{N} \alpha_j^k \left[ \langle \mu_{\mathbb{P}_t}, \mu_{\mathbb{P}_j} \rangle + \frac{1}{N^2} \sum_{k=1}^{N} \sum_{m=1}^{N} \langle \mu_{\mathbb{P}_k}, \mu_{\mathbb{P}_m} \rangle - \frac{1}{N} \sum_{k=1}^{N} \langle \mu_{\mathbb{P}_k}, \mu_{\mathbb{P}_j} \rangle - \frac{1}{N} \sum_{m=1}^{N} \langle \mu_{\mathbb{P}_m}, \mu_{\mathbb{P}_t} \rangle \right], \tag{11}$$

where the expression for $\alpha_j^k$ is given in the Appendix.

### 3.3 PROTOTYPE EMBEDDING

Let $\mathcal{N}$ be a neural network. Then the prototype embedding of probability measure $\nu$ can be given as,

$$\mathbf{u} = \mathbb{E}_{\mathbf{x}\sim\nu}\mathcal{N}(\mathbf{x}). \tag{12}$$

A suitable choice for $\mathcal{N}$ is the encoder part of a trained variational autoencoder. This encoder compresses all training probability distributions into a latent space, from which the original data can be fully reconstructed using latent samples. By taking the ensemble average of these latent samples, the embedding for each probability distribution is extracted. These probability embeddings are then fed into the score neural operator, which identifies the score function of the current probability distribution.

## 4 MODEL FORMULATION

### 4.1 SCORE NEURAL OPERATOR

We utilize NOMAD as the operator learning architecture to learn the score functions of probability distributions, primarily because it provides a continuous representation of the output function, which is essential for managing high-dimensional datasets such as images. In NOMAD, the encoder processes the vector representation $\mathbf{u}^\nu$ of each probability measure $\nu$, utilizing either equation 11 or equation 12. The decoder, on the other hand, processes the input $\mathbf{y} = [\mathbf{x}, t]$. The output from NOMAD is the score function of $\nu$, evaluated at $\mathbf{y}$. We generalize the objective function, as shown in equation 3 to accommodate multiple probability measures in the following form:

$$\min_\theta \mathbb{E}_{\nu\sim\mathcal{P}(\mathcal{X})}\mathbb{E}_{t\sim\mathcal{U}(0,T)}\lambda(t)\mathbb{E}_{\mathbf{x}(0)\sim\nu}\mathbb{E}_{\mathbf{x}(t)\sim p_{0t}(\mathbf{x}(t)|\mathbf{x}(0))}\left[\|\mathbf{s}_\theta(\mathbf{u}^\nu,\mathbf{x}(t),t) - \nabla_{\mathbf{x}(t)}\log p_{0t}(\mathbf{x}(t)\mid\mathbf{x}(0))\|_2^2\right]. \tag{13}$$

Minimizing equation 13 will derive the optimal $\mathbf{s}_\theta^{\text{opt}}(\mathbf{u}^\nu,\mathbf{x}(t),t)$ for any $\nu\sim\mathcal{P}(\mathcal{X})$. With the trained $\mathbf{s}_\theta^{\text{opt}}$, we can integrate the probability flow ODE equation 14 derived by Song et al. (2020) to generate samples for any $\nu\sim\mathcal{P}(\mathcal{X})$,

$$d\mathbf{x} = \left[\mathbf{f}(\mathbf{x},t) - \frac{1}{2}\nabla\cdot[\mathbf{G}(\mathbf{x},t)\mathbf{G}(\mathbf{x},t)^\mathrm{T}] - \frac{1}{2}\mathbf{G}(\mathbf{x},t)\mathbf{G}(\mathbf{x},t)^\mathrm{T}\mathbf{s}_\theta^{\text{opt}}(\mathbf{u}^\nu,\mathbf{x}(t),t)\right]dt. \tag{14}$$

### 4.2 LATENT SPACE SCORE NEURAL OPERATOR

(Vahdat et al., 2021), (Rombach et al., 2022) enhance the performance and speed of score-based generative models by conducting score matching in low-dimensional latent spaces. They use a variational autoencoder to compress the original data distribution into a low-dimensional latent space and perform the diffusion process there. Score matching in the latent space is significantly easier and faster than in the original image pixel space. Another advantage of utilizing the latent space is the wide mode coverage, which leads to a more expressive generated distribution. Inspired by their approach, we employ a VAE to map probability measures from the original pixel space $\mathcal{P}(\mathcal{X})$ to a latent space $\mathcal{Z}(\mathcal{Y})$ with Kullback-Leibler (KL) regularization. Subsequently, we train the score neural operator in this latent space.

Specifically, let $\mu\sim\mathcal{P}(\mathcal{X})$ be a probability measure. Recall that for a single distribution, the variational lower bound on negative data log-likelihood is given by,

$$\mathcal{L}_{\varphi,\phi}(\mathbf{x}) = \mathbb{E}_{q_\phi(\mathbf{z}|\mathbf{x})}\left[-\log p_\varphi(\mathbf{x}|\mathbf{z})\right] + \beta\mathbf{KL}\left(q_\varphi(\mathbf{z}|\mathbf{x})||\mathcal{N}(\mathbf{0},\mathbf{I})\right). \tag{15}$$

Equation 15 can be generalized to multiple probability distributions with the following optimization target,

$$\min_{\varphi,\phi}\mathcal{L}_{\varphi,\phi}^{\text{VAE}}(\beta) = \mathbb{E}_{\mu\sim\mathcal{P}(\mathcal{X})}\mathbb{E}_{\mathbf{x}\sim\mu}\mathbb{E}_{q_\phi(\mathbf{z}|\mathbf{x})}\left[-\log p_\varphi(\mathbf{x}|\mathbf{z})\right] + \beta\mathbf{KL}\left(q_\varphi(\mathbf{z}|\mathbf{x})||\mathcal{N}(\mathbf{0},\mathbf{I})\right). \tag{16}$$

We can conduct score matching in the latent space through,

$$\min_\theta\mathcal{L}_\theta^{\text{SGM}} = \mathbb{E}_{\nu\sim\mathcal{Z}(\mathcal{Y})}\mathbb{E}_{\mathbf{z}(0)\sim\nu}\mathbb{E}_{t\sim\mathcal{U}(0,1)}\lambda(t)$$
$$\mathbb{E}_{\mathbf{z}(t)\sim p_{0t}(\mathbf{z}(t)|\mathbf{z}(0))}\left[\left\|\mathbf{s}_\theta(\mathbf{u}^\nu,\mathbf{z}(t),t) - \nabla_{\mathbf{z}(t)}\log p_{0t}(\mathbf{z}(t)\mid\mathbf{z}(0))\right\|_2^2\right]. \tag{17}$$

The dependency on the distribution $\mu$ is replaced by its vectorized embedding $\mathbf{u}$ during computation, and we do not differentiate between them in the expressions. In equation 16 $\beta$ is a hyperparameter that balances the reconstruction error and the regularization error. Our training objective can be summarized as follows:

$$\min_{\phi,\varphi,\theta} \mathcal{L}^{\text{VAE}}_{\phi,\varphi}(\beta) + \gamma\mathcal{L}^{\text{SGM}}_{\theta}, \tag{18}$$

where $\gamma$ is a hyperparameter that balances the VAE loss and score-matching loss. When sampling from a test probability measure, we first compute its embedding $\mathbf{u}$ using equation 11 or equation 12. Next, we sample from a standard normal distribution and denoise back to the latent distribution using the predicted score function $\mathbf{s}^{\text{opt}}_{\theta}(\mathbf{u}, \mathbf{x}(t), t)$ and probability flow ODE equation 14. Finally, the generated samples are transformed back to the original pixel space through the decoder.

## 5 EXPERIMENTS

### 5.1 EVALUATION METRIC OF GENERATED SAMPLES

We discuss several metrics to assess the quality of samples generated by a generative model. Fréchet inception distance (FID) is a metric commonly used to evaluate the quality of generated images. To evaluate the quality of generated samples in low-dimensional space, we use the MMD (Maximum Mean Discrepancy) as the metric. The maximum mean discrepancy between two distribution $\mathbb{P}, \mathbb{Q}$ is an integral probability metric, given by Muandet et al. (2017),

$$\text{MMD}(\mathbb{P}, \mathbb{Q}, \mathcal{H}) = \sup_{\|f\|\leq 1}\left\{\int f(\mathbf{x})\mathbb{P}(\mathbf{x}) - \int f(\mathbf{y})\mathbb{Q}(\mathbf{y})\right\} = \|\mu_{\mathbb{P}} - \mu_{\mathbb{Q}}\|_{\mathcal{H}}. \tag{19}$$

Equation 19 is non-negative and lower bounded by 0 when $\mathbb{P} \overset{d}{=} \mathbb{Q}$. It can be estimated using empirical MMD,

$$\bar{\text{MMD}}(\mathbb{P}, \mathbb{Q}, \mathcal{H}) = \frac{1}{(m-1)m}\sum_{i=1}^{m}\sum_{j\neq i}^{m} k(\mathbf{x}_i, \mathbf{x}_j) + \frac{1}{(n-1)n}\sum_{i=1}^{n}\sum_{j\neq i}^{n} k(\mathbf{y}_i, \mathbf{y}_j) - \frac{2}{mn}\sum_{i=1}^{m}\sum_{j=1}^{n} k(\mathbf{x}_i, \mathbf{y}_j). \tag{20}$$

For grayscale images, such as those from the MNIST dataset, we employ a ResNet-18 architecture He et al. (2016) to train a digit classifier. We then utilize this trained classifier to assess the classification accuracy of digits generated from a test probability measure.

### 5.2 2D GAUSSIAN MIXTURE MODELS (GMMs)

For our 2D toy example, we utilize a family of Gaussian Mixture Models. Each model comprises four components, with centers positioned at the center of cells in a $6 \times 6$ lattice. All components follow a uniform distribution within the range of the cell they lie in. We define two panels within the lattice: the left panel consists of the first three columns, and the right panel includes the last three columns. For the training distributions, we randomly select two components from the same row in the left panel and two from the same column in the right panel, a configuration we denote as "left row right col." Similarly, we select two components from the same column in the left panel and two from the same row in the right panel, referred to as "left col right row." The testing distributions, however, are composed of combinations labeled as "left row right row." Given the low-dimensional nature of our data, we employ the MMD metric, as introduced in Section 5.1 to assess the quality of the generated samples. Each probability distribution is mapped to a RKHS using KME, and the probability embedding is evaluated using equation 11. The computation of inner products in RKHS is notably efficient in this low-dimensional setting. We generate 2000 training examples to train the score neural operator. The model is then used to predict the score functions for testing distributions, which feature patterns not present during the training phase. The results demonstrated in figure 1 shows that Score Neural Operator can generate high-quality samples from unseen datasets, comparable to those produced by individual score-based generative models.

### 5.3 $32\times32$ DOUBLE-DIGIT MINST

We developed an MNIST double-digit dataset, ranging from 00 to 99, by concatenating pairs of single digits from the original MNIST dataset. This process generated 100 unique probability distributions, from which we selected 70 for training the Score Neural Operator and 30 for testing.

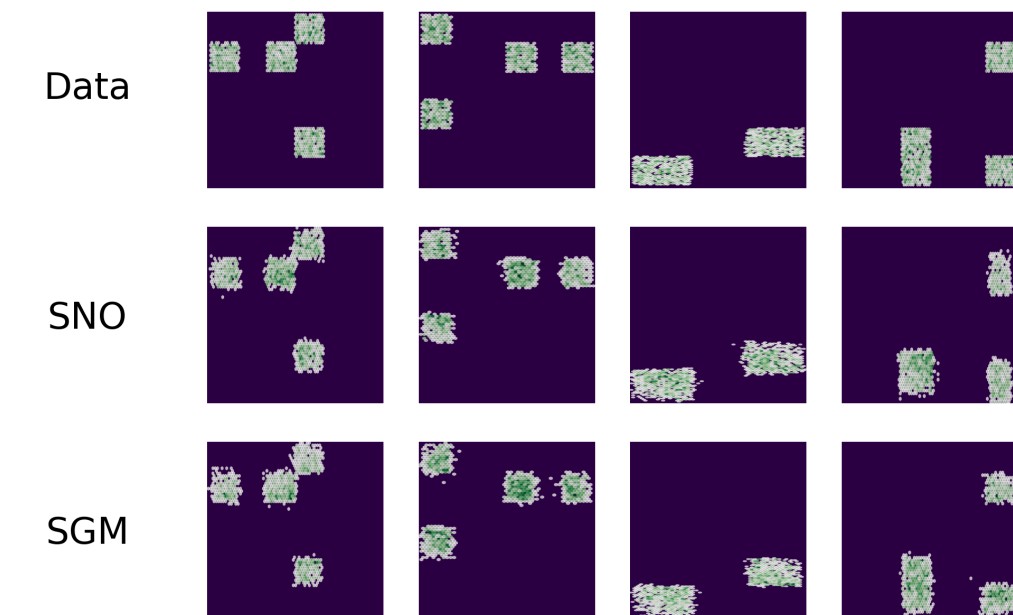

Figure 1: We examine four 2D Gaussian Mixture Model (GMM) distributions: the first two are derived from the training sets, and the last two are from the testing sets. To generate samples for these distributions, we utilized four score-based generative models (SGMs) and a Score Neural Operator (SNO). The first row displays raw samples from all four distributions. The second row presents samples generated using the SNO, while the third row shows samples generated by the four distinct SGMs. The maximum mean discrepancy (MMD) between samples generated by the SGMs and the original dataset are 0.0070, 0.0053, 0.0079, and 0.0056 (from left to right). For the SNO, the MMD values are 0.0054, 0.0064, 0.0081, and 0.0148, respectively. These results indicate that the SNO produces samples of comparable quality to those generated by individual SGM models.

Performing score-matching directly in a 1024-dimensional pixel space proved time-consuming and susceptible to overfitting. To address this, we implemented the latent space technique outlined in section 4.2, which involves training a Variational Autoencoder (VAE) and conducting low-dimensional score-matching in an end-to-end manner using equation 18. We executed multiple experiments under varied settings to assess the classification accuracy on both training and testing datasets. The results are summarized in Table 1. Our experiments revealed that end-to-end training faces convergence challenges when KMEs are computed in a dynamically changing latent space. However, these challenges are mitigated when prototype embeddings are utilized. By computing KMEs in the original pixel space and employing fixed probability embeddings **u** for end-to-end training, we achieved rapid convergence. We also observed that the stability of the training process is significantly impacted when the Variational Autoencoder and score-matching are trained separately. This approach resulted in considerable variability in training and testing classification accuracies, which were highly dependent on the random seed used. Figure 4 and Figure 5 show the generated training and testing digits after the training has fully converged. The results indicate that the Score Neural Operator can generate high-quality images for both the training and testing sets, demonstrating its ability to accurately predict the score functions of unseen datasets.

To demonstrate our model's ability to accurately predict score functions for distributions not included in the training set, we employed a conditional score model as a baseline. This model also incorporates a probability embedding as input to identify specific probability distributions. Unlike the embeddings in our Score Neural Operator, which are interconnected across different distributions, the embeddings in the conditional score model resemble isolated one-hot encodings, lacking inherent generalization capabilities. To verify this, we trained the conditional score model on 70 training digits over 10,000 epochs, followed by finetuning on 30 testing digits. Both training and testing accuracies were continuously monitored (see Figure 2). The results indicate that while the

Table 1: Comparative Performance of Score Neural Operator Configurations: We compare the performance of our model in latent and pixel spaces, using various embedding methods (Prototype, KME, and Conditional) and sample sizes.

| Exps | Space | # Samples | Embedding | Train Acc. | Test Acc. |
|------|-------|-----------|-----------|------------|-----------|
| 1 | Latent | 2000 | Prototype | 89.5% | 84.2% |
| 2 | Latent | 2000 | KME | 88.0% | 80.0% |
| 3 | Latent | 2000 | Conditional | 87.2% | 0.9% |
| 4 | Pixel | 2000 | KME | 94.8% | 61.1 % |
| 5 | Pixel | 2000 | Prototype | 95.2% | 60.1 % |

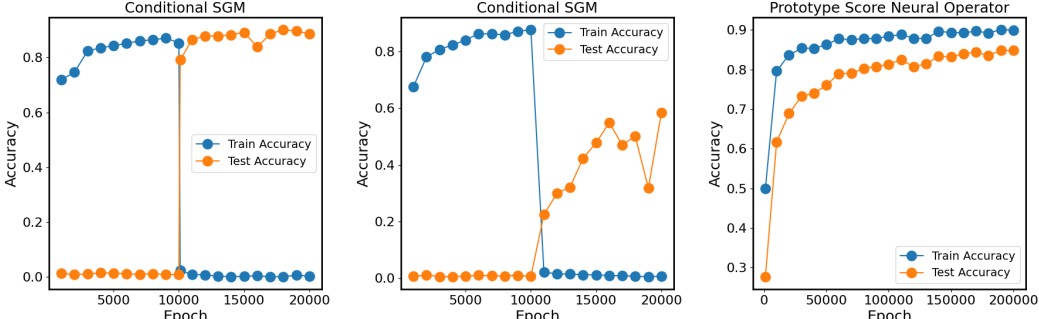

Figure 2: Classification accuracy for training and testing sets over the course of training using a conditional score-based generative model (left and middle) and a Score Neural Operator (right). For the conditional model, training initially involves 10,000 epochs on 70 training probability distributions, each containing 2,000 samples, followed by fine-tuning for another 10,000 epochs on 30 testing distributions. The left sub-figure displays results with 2,000 samples per testing distribution, while the middle sub-figure presents results with only 1 sample per testing distribution. The right sub-figure shows the results of the Score Neural Operator using the prototype embedding method, trained on 70 training digits with 2,000 samples each. The results indicate that the conditional score-based generative model can only perform well on either the training set or the testing set, but not both simultaneously. In contrast, the Score Neural Operator demonstrates robust performance on both sets, despite being trained solely on the training set.

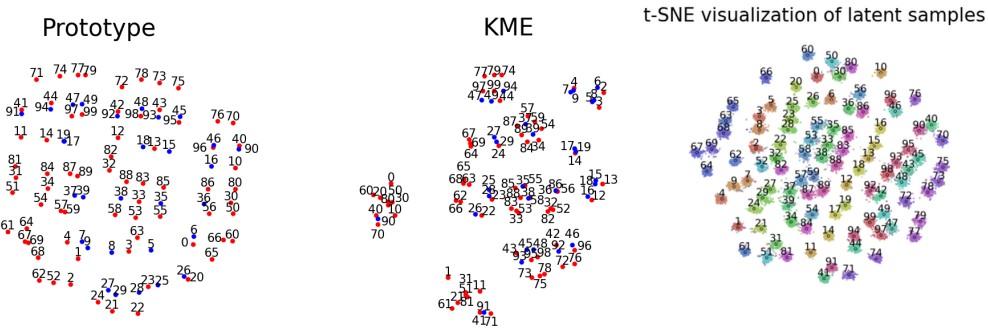

Figure 3: t-SNE visualization of the probability embedding **u** with Prototype embedding (left) and KME embedding (middle) for 70 training distributions (red) and 30 testing distributions (blue). The right sub-figure displays the t-SNE visualization of latent samples from all 100 distributions. The results indicate that all 100 distributions are well-separated in the latent space, with distributions sharing many similar features positioned close to each other.

score functions for new distributions can be efficiently trained using well-pretrained weights, the model does not exhibit generalization proficiency in predicting score functions for unseen datasets.

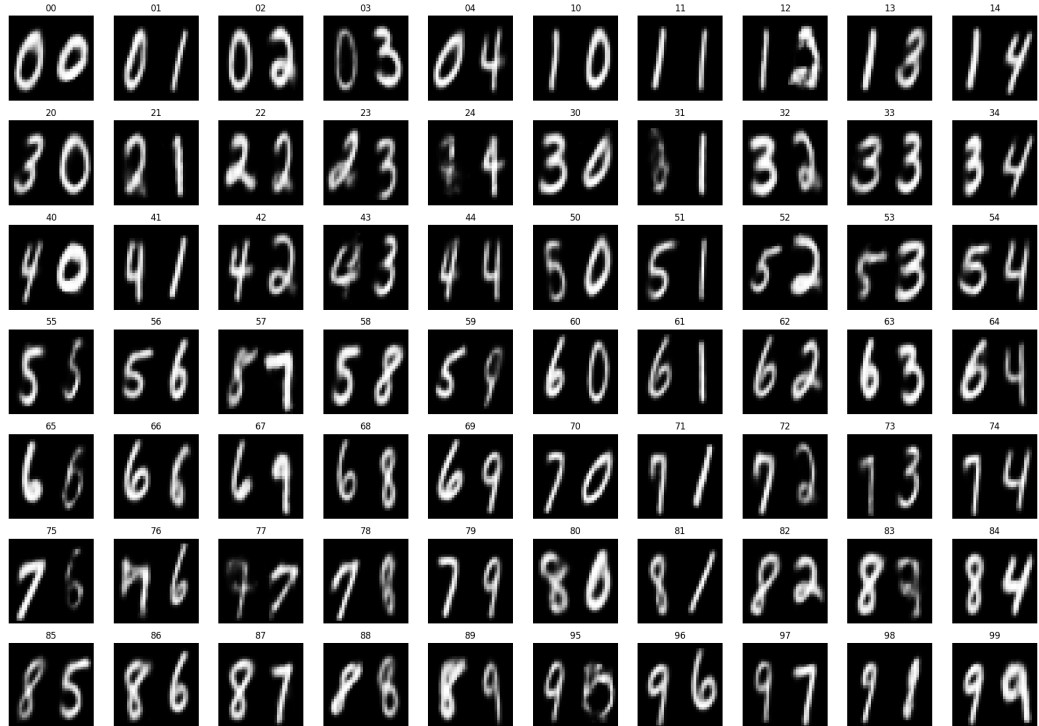

Figure 4: The Score Neural Operator is trained on 70 training distributions, each containing 2,000 samples. Subsequently, the model generates an image for each of the 70 training distributions. The results demonstrate that the model is capable of producing high-quality images across multiple training datasets.

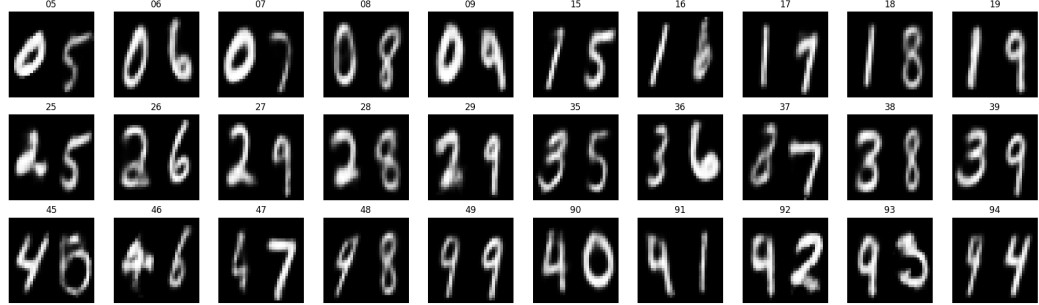

Figure 5: The Score Neural Operator is trained on 70 training distributions, each containing 2,000 samples. It subsequently generates images for each of the 30 testing distributions, also containing 2,000 samples each, to evaluate the probability embedding **u**. The results illustrate that the model is capable of producing high-quality images for datasets it has never seen before, without the need for retraining.

## 5.4 APPLICATIONS TO FEW SHOT LEARNING

Utilizing a score-based generative model to learn the distribution of a single image typically results in a delta function centered on that image, with the generated sample being essentially the original image plus some noise. However, when our trained score neural operator is applied to an unseen image, it is capable of generating a variety of distinct images that adhere to the same underlying data distribution. As illustrated in figure 6, this approach can generate distinct images from only a single input image from a new distribution, achieving approximately 74% classification accuracy. This demonstrates a significant potential for enhancing few-shot learning capabilities. In contrast, a

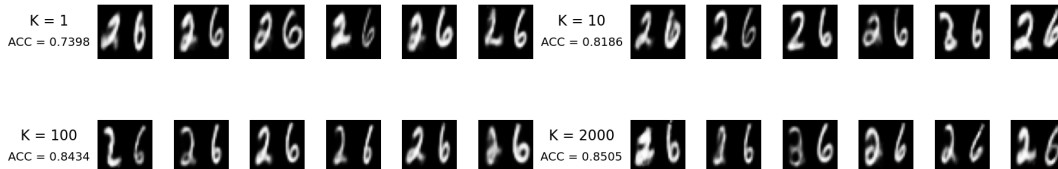

Figure 6: The Score Neural Operator is trained on 70 training distributions, each containing 2,000 samples. It subsequently generates images for 30 testing distributions using a probability embedding **u** computed from only a few samples, denoted as K. We use 'ACC' to represent the averaged classification accuracy of 1,000 generated images for each of the 30 testing distributions. The ACC values are 0.7398, 0.8186, 0.8434, and 0.8505 for K=1, 10, 100, and 2000, respectively. The test digit '26' is included to illustrate the quality of the generated images. The results indicate that the model can produce diverse samples from unseen probability distributions using just one testing sample, highlighting its potential for few-shot learning applications.

conditional score-based generative model cannot generalize to unseen test distributions once trained. Even with fine-tuning using a single sample, it can only learn the delta function and reproduce the test image (see the right part of Figure 2).

## 6 DISCUSSION

**Summary.** Our study introduces the Score Neural Operator, a novel generative modeling framework that learns to map multiple probability distributions to their corresponding score functions. This approach enables generalization to unseen distributions without retraining, marking a significant advancement in the field. We have demonstrated its effectiveness on both low-dimensional Gaussian Mixture Models and high-dimensional MNIST double-digit data, showcasing strong generalization performance. Our implementation of latent space techniques for score matching has proven effective in handling high-dimensional data, improving both training efficiency and sample quality. Furthermore, we've shown the potential of our method for few-shot learning applications, where a single example from a new distribution can generate diverse, high-quality samples.

**Limitations.** Despite these promising results, the Score Neural Operator has limitations. Its performance may degrade when faced with distributions significantly different from those seen during training. The computational cost of training on a large number of diverse distributions could become prohibitive for very large-scale applications. Additionally, while our method shows improved generalization compared to traditional approaches, it still requires a substantial amount of training data to learn the underlying operator mapping. These limitations highlight the need for further research to enhance the model's scalability and generalization capabilities.

**Future Work.** Future work could explore several promising directions. Investigating the theoretical foundations of the Score Neural Operator's generalization capabilities could provide insights into its performance and guide future improvements. Integrating this approach with other advanced generative modeling techniques, such as transformer architectures or neural ordinary differential equations, could further enhance its capabilities. Extending the model to handle conditional generation tasks would greatly increase its practical utility. Additionally, exploring more efficient training algorithms or adaptive learning strategies could address the scalability concerns identified in our current implementation.

**Potential Societal Impact.** The potential societal impact of this work is significant, with applications ranging from improved data augmentation techniques in healthcare imaging to more adaptable AI systems in rapidly changing environments. However, as with any advanced generative model, there is potential for misuse, such as in the creation of deepfakes or other synthetic media for malicious purposes. It is crucial that future development and deployment of these technologies be accompanied by robust ethical guidelines and safeguards.

## REPRODUCIBILITY STATEMENT

The code used to carry out all experiments is provided as a supplementary material for this submission. If the paper is accepted, we plan on making our entire code-base publicly available on GitHub.

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

# A   APPENDIX

## A.1   IMPLEMENTATION DETAILS

**NOMAD.**   Our NOMAD architecture consists of three MLPs: a branch layer, a trunk layer, and an output layer. Each of these MLPs is 7 layers deep, with 500 neurons per layer and GELU activation functions. The NOMAD is employed for running 2D-GMMs experiments.

**VAE.**   Given the simplicity of the MNIST double-digit dataset, we utilized a 3-layer deep MLP with 512 neurons per layer and ReLU activation functions for both the encoder and decoder components.

**ScoreNet.**   For conducting score-matching in the original pixel space, we used a network comprising 3 down-sampling blocks and 3 up-sampling blocks. Each block includes a convolutional layer and a max-pooling layer, with LogSigmoid activation functions. For the score network in the latent space, we employed 2 down-sampling blocks and 2 up-sampling blocks, each consisting of a 2-layer deep MLP with LogSigmoid activation functions.

## A.2   EXPERIMENT SETUP

**Hyperparameters.**   For the 2D-GMMs experiments, we used the Variance Exploding Stochastic Differential Equation (VESDE) and an ODE sampler with the hyperparameter $\sigma = 25.0$. For the MNIST double-digit dataset, we employed the Variance Preserving Stochastic Differential Equation (VPSDE) and used the Euler-Maruyama method to solve the reverse diffusion process, with

$\beta_{\max} = 20$ and $\beta_{\min} = 0.1$. Additionally, we incorporated Fourier Feature Embedding into the NO-MAD architecture for the 2D-GMMs experiments, where the random Gaussian matrix was sampled from $\mathcal{N}(0, 10^2)$. The parameter $\gamma$ is set to 1 in equation 18, and the parameter $\beta$ is set to 2048 in equation 16.

**Environment Setup.** Our experiments were conducted on an NVIDIA GeForce RTX 3070 GPU. The software environment included JAX version 0.4.33, JAXLIB version 0.4.33, PyTorch version 2.0.1+cu117, CUDA version 11.5, CUDNN version 8.9.2, and Python 3.10.

### A.3 KMEs COEFFICIENTS

Let $\hat{C}$ be the operator on the RKHS. First we scale each KME by the empirical mean of KMEs over N_Train $= 5000$ training examples $\bar{\mu}_{\mathbb{P}_i} = \mu_{\mathbb{P}_i} - \bar{\mu}_{\mathbb{P}}$, where $\bar{\mu}_{\mathbb{P}} = \frac{1}{N} \sum_{j=1}^{N} \mu_{\mathbb{P}_j}$. We consider $\mathbf{v}$ from a finite subspace spanned by a linear combination of $\bar{\mu}_{\mathbb{P}_i}$, $\mathbf{v} = \sum_{i=1}^{N} \alpha_i \bar{\mu}_{\mathbb{P}_i}$. Then,

$$
\begin{aligned}
\hat{C}\mathbf{v} &= \frac{1}{N} \sum_{i=1}^{N} (\bar{\mu}_{\mathbb{P}_i} \otimes \bar{\mu}_{\mathbb{P}_i})\mathbf{v} = \frac{1}{N} \sum_{i=1}^{N} <\mathbf{v}, \bar{\mu}_{\mathbb{P}_i}> \bar{\mu}_{\mathbb{P}_i} = \frac{1}{N} \sum_{i=1}^{N} \sum_{k=1}^{N} \alpha_k <\bar{\mu}_{\mathbb{P}_k}, \bar{\mu}_{\mathbb{P}_i}> \bar{\mu}_{\mathbb{P}_i} \\
&= \lambda\mathbf{v} = \lambda \sum_{i=1}^{N} \alpha_i \bar{\mu}_{\mathbb{P}_i},
\end{aligned}
\tag{21}
$$

from which we get

$$
\sum_{k=1}^{N} \alpha_k <\bar{\mu}_{\mathbb{P}_k}, \bar{\mu}_{\mathbb{P}_i}> -N\lambda\alpha_i = 0, \quad i = 1, 2, \cdots, N
\tag{22}
$$

$$
(\mathbf{M} - N\lambda\mathbf{I})\boldsymbol{\alpha} = \mathbf{0},
\tag{23}
$$

where $\mathbf{M}_{ij} = <\bar{\mu}_{\mathbb{P}_i}, \bar{\mu}_{\mathbb{P}_j}>$. Note that

$$
\begin{aligned}
<\bar{\mu}_{\mathbb{P}_i}, \bar{\mu}_{\mathbb{P}_j}> &= \left\langle \mu_{\mathbb{P}_i} - \frac{1}{N} \sum_{k=1}^{N} \mu_{\mathbb{P}_k}, \mu_{\mathbb{P}_j} - \frac{1}{N} \sum_{m=1}^{N} \mu_{\mathbb{P}_m} \right\rangle \\
&= \langle \mu_{\mathbb{P}_i}, \mu_{\mathbb{P}_j} \rangle + \frac{1}{N^2} \sum_{k=1}^{N} \sum_{m=1}^{N} \langle \mu_{\mathbb{P}_k}, \mu_{\mathbb{P}_m} \rangle - \frac{1}{N} \sum_{k=1}^{N} \langle \mu_{\mathbb{P}_k}, \mu_{\mathbb{P}_j} \rangle - \frac{1}{N} \sum_{m=1}^{N} \langle \mu_{\mathbb{P}_m}, \mu_{\mathbb{P}_i} \rangle.
\end{aligned}
\tag{24}
$$

It's clear that $\boldsymbol{\alpha}$ is eigenvector of $\mathbf{M} - N\lambda\mathbf{I}$ with eigenvalue $N\lambda$. Hence, we can solve the first $N_x$ eigenvectors $\boldsymbol{\alpha}^1, \cdots, \boldsymbol{\alpha}^{N_x}$ and use them to compute the first $N_x$ eigenvectors $\mathbf{v}$ for $\hat{C}$ the spans of which covers enough information of $\{\bar{\mu}_{\mathbb{P}_i}\}$. Then for $\mathbb{P}_i$, the input for the encoder of NOMAD can be evaluated as $[<\mathbf{v}_1, \bar{\mu}_{\mathbb{P}_i}>, \cdots, <\mathbf{v}_{N_x}, \bar{\mu}_{\mathbb{P}_i}>]$. Note that $<\mathbf{v}_k, \bar{\mu}_{\mathbb{P}_i}> = \sum_{j=1}^{N} \alpha_j^k <\bar{\mu}_{\mathbb{P}_j}, \bar{\mu}_{\mathbb{P}_i}>$. For a testing distribution $\mathbb{P}_t$, we need to compute

$$
\begin{aligned}
<\mathbf{v}_k, \bar{\mu}_{\mathbb{P}_t}> &= \sum_{j=1}^{N} \alpha_j^k <\bar{\mu}_{\mathbb{P}_j}, \bar{\mu}_{\mathbb{P}_t}> = \sum_{j=1}^{N} \alpha_j^k \left\langle \mu_{\mathbb{P}_t} - \frac{1}{N} \sum_{k=1}^{N} \mu_{\mathbb{P}_k}, \mu_{\mathbb{P}_j} - \frac{1}{N} \sum_{m=1}^{N} \mu_{\mathbb{P}_m} \right\rangle \\
&= \sum_{j=1}^{N} \alpha_j^k \left[ \langle \mu_{\mathbb{P}_t}, \mu_{\mathbb{P}_j} \rangle + \frac{1}{N^2} \sum_{k=1}^{N} \sum_{m=1}^{N} \langle \mu_{\mathbb{P}_k}, \mu_{\mathbb{P}_m} \rangle - \frac{1}{N} \sum_{k=1}^{N} \langle \mu_{\mathbb{P}_k}, \mu_{\mathbb{P}_j} \rangle - \frac{1}{N} \sum_{m=1}^{N} \langle \mu_{\mathbb{P}_m}, \mu_{\mathbb{P}_t} \rangle \right].
\end{aligned}
\tag{25}
$$

