# OpenReview forum: "SCORE NEURAL OPERATOR: A GENERATIVE MODEL FOR LEARNING AND GENERALIZING ACROSS MULTIPLE PROBABILITY DISTRIBUTIONS"
_ICLR.cc/2025/Conference — ICLR 2025 Conference Withdrawn Submission_

### Official Review · Reviewer_nziV · 2024-10-28

**Soundness:** 2
**Presentation:** 1
**Contribution:** 2
**Rating:** 1
**Confidence:** 4

**Summary:**

The authors tackle the problem of generating sampling from unseen distributions using score-based approaches at inference. They propose to use the encoder of the VAE to embed specific samples. Then they represent the embedding of specific dataset/data-distribution as the average of the sample embeddings for this dataset. Then conditional on the distribution embedding from a collection of samples on a sampled distribution, they generate new samples of that distribution with score matching. The goal is to do some kind of meta-learning, i.e.: embed a small dataset and then have their model generate more samples of that distribution (unseen in the training, so OOD) in a zero-shot manner. Sadly, the abstract and title makes it very difficult to understand. Their specific choice of average embedding (called prototype) gives them the best generalization on toy experiments (2D Gaussian and paired MNIST digits).

**Strengths:**

Simple way to get zero-shot generation on new distributions. Has potential for interesting applications (not tackled in this paper, which only deal with toy problems).

**Weaknesses:**

The paper is extremely poorly written. It makes it extra hard to understand. Definitions are unclear and change over the course of the paper. Its a guessing game to understand what is what. I had an horrible time trying to understand the paper. Readers who are not reviewers will clearly just skip this paper given how badly written it is.

The paper title and abstract needs refactoring to be comprehensible when reading for the first time. The paper needs to actually define and explains things properly.

Experiments are nice but very toy-ish, 2D Gaussians and pairs of MNIST digits. You need something like ImageNet with some classes never seen. CIFAR-100 also could be a start, remove like 5 classes from it for test time. This is whole idea of out-of-distribution generalization. There are existing benchmarks for that, but its more for classification. But you could just reuse them for generation.See Domainbed, https://github.com/facebookresearch/DomainBed, you train on some domains and not on others. This would be perfect to test your model. Also it has been shown that diffusion models can do zero-shot classification (https://arxiv.org/abs/2303.16203), you could even test your approach on the actual DomainBed challenge for classification using your score-model! Sky is the limit, but right now the paper is not ambitious enough and its not looking into interesting applications. With a bit more ambition, this could be a very interesting paper.

It needs more extensive literature review for meta-learning approaches.

**Questions:**

Line 47: I would recommend adding "and by reversing the stochastic differential equation" or something like this. Very few people using Langevin anymore.

Line 142: Right now there is background and new upcoming parts in the background section. Its fine, but it would be nice to more clearly separate prior from new work. Adding let say a small heading like a \paragraph or subsection before new information would be nice to clearly separate new from old.

Line 212: μ_P goes to μˆ_Pi with no explanation (what is Pi?), then to P_t (what is t?), then to the k-th element of u (not μ, idk what this is). The equation looks like it contains MMD, but it comes out of nowhere. It refers to Appendix. But not where in the Appendix. I found it in A.3, but this really doesn't help me understand better. The u is also written differently then in equation 12. Its badly presented and confusing.

Line 218: N is a very weird choice for neural network notation. I suggest changing it since N is also for N(0,1). Now v is the measure, before it was u_P and u_Q. I have no idea why it changed again.

Line 235: Would be great to define u^v in line 220... It has never been defined, but I can guess that it is just N(x).

Line 232: NOMAD is only mentioned in the Related Work, which is an optional section. NOMAD needs to be in the background.

Line 249: Remove the parenthesis of the citations.

Line 255: Would be nice to say that these prior work use VAE with KL and following this, you do it too because otherwise a reader unfamiliar with StableDiffusion might be wondering where did the KL regularization came from and if you invented it.

Line 277: I would like to remind you that u^v has never been defined, we can only guess what it is. And now equation 14 and 17 have u^v, but the text mention u! What is happening here! I finally understood that u^v is equation 12, its just a typo, its missing the v. Same thing for Equation 11, which incorrectly defined, since it doesn't even have a v in it. This finally make sense. This makes me understand equation 17.

Line 298: Doesn't there exists FID networks for MNIST in the literature? Just wondering. But the approach of training your own is fine.

---

### Official Review · Reviewer_4jAm · 2024-10-31

**Soundness:** 3
**Presentation:** 3
**Contribution:** 2
**Rating:** 5
**Confidence:** 3

**Summary:**

The authors propose a Score neural operator, that learns to model the score in a distribution-agnostic way, thereby potentially allowing for the score of unseen distributions to be estimated with very few training samples.

**Strengths:**

The proposed formulation is novel and has the scope to be applicable to multiple scenarios if successful. Although my expertise is not in the space of neural operators, the theoretical formulation presented is well established and correct, to the best of my understanding.

**Weaknesses:**

The paper has a few weaknesses that could be addressed:

-  Writing: In general, the writing could use some improvement. For example, citations are poorly done, with incorrect usage of \citep and \citet (L34-42,L59,L46-50, etc.) especially in places where the authors themselves write a paper’s author names, followed by \citet. (E.g., L86,89,91). These could be fixed. I see similar issues with equations (L129).

-  Math: Sometimes, the notations are defined when they first appear, and this could cause confusion. For example, the definition of $\mathbf{u}$ in L210 is not accompanied by a definition of what it represents. Similarly, the use of $\mathbf{G}$ in Eqn. 14, or $\nu$ in Eqn 12. In most cases, this is described eventually, say half a page later, but this does not yield a good reading experience and could be fixed.

- Experimental validation: Though the proposed idea and methodology seems promising, the experiments, in my opinion, don’t live up to the claim. Unless I am misunderstanding, It would be best to either make the claims more tamer, or have experiments the validate the existing claim that the SNO would be able to work few-shot with any data distribution. The experiments on 2-digit MNIST with a hold out set do not seem convincing. Even if they chose to work with lower resolution images, it would be more appealing to see actual generalization across datasets. For example, how about using SVHN (greyscale) and MNIST. Both being digit datasets, the task would be manageably hard, but possible also able to justify that few-shot alignment to a brand new dataset is indeed possible.

**Questions:**

Please see Weaknesses

---

### Official Review · Reviewer_QdtG · 2024-11-01

**Soundness:** 2
**Presentation:** 1
**Contribution:** 2
**Rating:** 3
**Confidence:** 3

**Summary:**

This paper introduces the score neural operator, an architecture that learns to map multiple probability distributions to their score functions, enabling generalization to unseen distributions. The ideas are developed in the latent space and are examined on toy and small datasets such as Gaussian Mixture Models or 1024-dimensional MNIST.

**Strengths:**

The problem of learning a shared generative model across different data distributions is a relevant and important problem with significant interest from the community.

The idea of using neural operators for this task seems interesting.

**Weaknesses:**

- The experiments are conducted in toy and small datasets such as Gaussian Mixture Models or 1024-dimensional MNIST. From my experience working with these datasets, it is hard to expect that findings in these datasets naturally generalize to real-world datasets.

- The idea of learning one generative model across different data distributions was originally discussed in "Distribution Augmentation for Generative Modeling" by Jun et al. ICML 2020. This paper shows that one can learn one generative across different distributions by providing a simple input conditioning that specifies the target distribution. This idea has been explored in diffusion modeling literature as well (for example see "Elucidating the Design Space of Diffusion-Based Generative Models" by Karras et al. NeurIP 2022). This submission does not discuss the idea of specifying the data distribution via simple embedding. It also does not discuss why a simple approach like passing a data distribution embedding would enable learning a diffusion model across different distributions.

- The paper is hard to follow. Section 3.2 suddenly introduces many different concepts without explaining how they fit into the bigger picture.

- Equation 18 trains the VAE and SGM prior jointly. However, the KL is evaluated with respect to a standard Normal prior in Eq. 16 while the prior is set to be a diffusion model.

**Questions:**

Please use \citet or \citep appropriately to avoid sentences like: "Song and Ermon Song & Ermon (2019) introduced .... "


What are the exact inputs and outputs to the score neural operator? Why does the neural operator generalize to the new score function?

---

### Official Review · Reviewer_sAY6 · 2024-11-01

**Soundness:** 2
**Presentation:** 1
**Contribution:** 2
**Rating:** 5
**Confidence:** 2

**Summary:**

This paper proposes the Score Neural Operator (SNO) algorithm, addressing the limitations of existing generative models that often rely on overly simplified distributions. SNO first uses a Variational Autoencoder (VAE) to map data into a latent space and subsequently applies score matching to shift the distribution toward a Gaussian. The algorithm demonstrates effective performance on toy examples, such as MNIST and Gaussian Mixture Models (GMMs).”

**Strengths:**

A notable strength of this approach is its similarity to Latent Diffusion, enabling the simultaneous training of the VAE and SNO. This joint training allows for a meaningful generation schema even with limited data, making it effective in scenarios where data is sparse. The use of score matching in the latent space further enhances the quality of generated samples, leveraging the efficiency of learning in a compressed representation.

**Weaknesses:**

Discrepancy Between Claims and Experimental Evidence:
- While the paper claims that SNO can improve performance across various distributions, the chosen datasets—GMM and MNIST—are actually toy dataset. These limited datasets are insufficient to convincingly demonstrate the model’s effectiveness for handling complex and high-dimensional distributions. If SNO’s strength lies in handling diverse and complex data, additional experiments with richer and more challenging datasets are necessary to substantiate this claim.

**Questions:**

1. What advantages does SNO offer over existing baselines in practical applications?
- For few-shot generation tasks, using models like StyleGAN with fine-tuning can yield high-quality results with less complexity. SNO does not clearly demonstrate an advantage over these existing models in terms of efficiency or output quality for few-shot tasks.
2. Is SNO truly scalable and beneficial for large datasets?
- If SNO aims to generalize across complex distributions, especially in large-scale benchmarks, it remains unclear whether its assumptions hold or if the model would perform meaningfully better than traditional score-based models on large, complex datasets. Without evidence of SNO’s scalability and robustness to diverse data, the practical benefits seem limited.
3. How does it differ from Latent Diffusion Model?
- It seems that it is lot similar to LDM. LDM also moves the image into latent space, then conducts diffusion process.

---

### Official Review · Reviewer_ypDA · 2024-11-04

**Soundness:** 3
**Presentation:** 1
**Contribution:** 2
**Rating:** 5
**Confidence:** 3

**Summary:**

The authors propose an improved score based model, capable of generalizing to unseen continuous distributions that are given as an input at test time. The distributions have a special representation and the model uses their continuous representation to calculate the score function. The method is demonstrated for several examples such as 2d toy examples, MNIST double digits. They also demonstrate using a single example or few of them to construct the distribution representation and get reasonable results.

**Strengths:**

1. The paper ambitiously addresses the challenge of generalizing score-based generative models across multiple distributions, relevant for few-shot learning and adaptive AI. Deriving a new model is refreshing and should be appreciated.

2. It introduces a mathematical framework using NOMAD for continuous distribution mapping, combining operator learning with generative modeling. This is a good idea to my taste, and a non-trivial contribution.

3. The use of KME and PCA for embedding distributions is quite elegant.

4. The authors demonstrate their method for latent space models as well. This is useful and important.

**Weaknesses:**

## 1) Readability
Usually one doesn't mark the first weakness on the list to be readability, but in this caseI find it crucial. I was having a very hard time to read this paper and follow the derivations.
1. First the structure is hard to follow. I would expect starting from high level and then get in to low level derivations. I don't mean the intro as the high level, but for example in section 4, I would expect to see clearly what is the input, output and goal.
2. I think some simple examples are also needed for inputs and expected outputs.
3. A figure describing the method is also missing.
4. In the background I think a brief explanation about NOMAD is needed.
5. Taking Figure 1 as a test case: It took me unreasonable amount of time to understand it. The related text in sec 5.2 is confusing and almost cryptic. It is not clear what is Gaussian there, no motivation for this very specific choice of data. No visual explanation in the figure itself. The color coding is not clear to me. The caption says "the first two are derived from the training sets, and the last two are from the testing sets." does that mean the left two and right two? If so why not putting some lines making a table with titles? Also if that is the case, I don't understand why the rightmost column has examples that are "left col right row" which is a training characteristic, not test.

## 2) Comparing to conditional generative models
1. Simply defining the goal, I did not see any definition that makes the premise of this work actually different from a conditional generative model. Conditional generative models in many cases get conditions that they were never trained on. They produce samples from a conditional distribution that I think can be any one of the examples in the experiments. In general I think that explaining why the proposed method is different/better than conditional models should be the most important claim to justify in this paper.
2. The authors do use a baseline they refer to as a conditional model. To my understanding it is something else. A model that is fine-tuned on the unseen distributions. First, this is not what a conditional model is- a conditional model gets a condition along with its input. Moreover, comparing to fine-tuned models implemented by the authors is always problematic. We don't have any details about how many iterations, what was the procedure, what weights trained etc. Without pointing any finger to any.one, usually authors don't have a huge motivation to make a baseline work better and fine-tuning is very tricky.
3. Mathematically, while I appreciate the interesting derivation, as far as I see, the objective boils down to a standard conditional setup. eq.13 is eq.3 with an additional conditioned term, sampled from a distribution of conditions. $s_\theta(u^v,z(t),t)$. I think any condition can be framed as a representation of a distribution. If the novelty is in the specific representation, then the premise should have been around it and indicating differences from the representation used in NOMAD. If not, then the contribution of the paper is combining NOMAD with conditional score models.
4. l503: "conditional score-based generative model cannot generalize to unseen test distributions once trained." I don't agree with that statement. While it might be true for discrete class-conditioning, it is not true in most other cases. Conditioning can be done for example on images in various ways and of course can be applied with an unseen image. Text-to-image models are another example for conditioning that generalizes to not necessarily seen conditions.

## 3) Experiments
1. In general experiments are done on very low-scale examples and very few. While I wouldn't expect generating HD videos with a novel method, I do think that more evidence for its capabilities is needed.
2. The toy example experiment is very specific and complicated. Why not choosing simple distributions? Like spirals or just Gaussians, as used in many generative models works? Seeing such examples raises the concern whether they were carefully chosen.
3. I'm not sure about the few-shot learning part. I think the fact that it is possible to use a single example actually very strongly shows that this is equivalent to a conditional model. If it really adhered accurately to this distribution it should, in fact, make a Dirac Delta over the given example and generate only that. Thinking about it as a condition also makes this application less impressive, as single images as conditions are common.

**Questions:**

It is very possible that I am missing some crucial details (maybe due to the readability issues). If that is the case, I will be more than happy to raise the rating. Otherwise, it seems to me that the paper is sort of reinventing conditional models, and does so very poorly written. With that said, I do find a contribution in the combination of representing continuous distributions and using them as a condition for score-based models. Currently that is not the story told by the authors. If it were, then I'm not sure, but perhaps It could be a reasonable contribution, subject to pin pointing exactly what existed before and what techniques that make it possible are novel.

---

### Note · Authors · 2024-11-20

**Comment:**

Thank you to all the reviewers for your valuable comments. It appears that many reviewers found our paper difficult to understand. We will revise our paper to include clearer definitions and illustrations of our models and conduct experiments on more advanced datasets.

**Withdrawal Confirmation:**

I have read and agree with the venue's withdrawal policy on behalf of myself and my co-authors.